

# An Improvement of the One-dimensional Ocean Wave Description based on SWIM Observations

Yihui Wang[1,2,3], Xingou Xu[2,3]

[1] Key Laboratory of Space Ocean Remote Sensing and Application, Ministry of Natural Resource, P. R. China

[2] Key Laboratory of Microwave Remote Sensing, National Space Science Center, Chinese Academy of Sciences, Beijing, 100190, China

[3] School of Electronic, Electrical and Communication Engineering, University of Chinese Academy of Sciences, Beijing, 100049, China

*Correspondence to*: Xingou Xu(xuxingou@mirslab.cn)

**Abstract.** The one-dimensional ocean wave spectra (1D spectra) describing the total energy of the ocean waves are vital for providing ocean surface roughness information in remote sensing simulations. Most existing wave spectrum models deviate from real ocean surface descriptions due to limitations of observation methods and approximation in theories applied to generating them. In this research, the widely applied Goda and Elfouhaily spectra in their 1-D form are compared with the remote sensing products from the Surface Waves Investigation and Monitoring instrument (SWIM) on-board the China France

Oceanography Satellite (CFOSAT). Differences between models and the measurements are addressed, then the causes are analyzed and concluded in terms of sea states. Then, a Combined spectrum (C spectrum) considering varied sea states is proposed as a closer model to the observations of the real sea, where parameterization of the spectral peak enhancement factor ($\gamma$) is achieved by the inverse wave age and wave steepness for multiple sea states. Then the specific values of the all-state sea are obtained from SWIM observations. The validation of the C spectrum is achieved by comparisons with SWIM

measurements not utilized during the model establishment, and with buoy measurements. The difference index (DI) and the R-squared ($R^2$), are calculated for evaluation of the results, indicating that the C spectrum demonstrates closer fitting to the SWIM and buoy measurements than both Goda and Elfouhaily spectra. The DI and $R^2$ for the C spectrum are compared to the Goda spectrum, which is closer to SWIM measurements than E spectrum, and values are 0.780 and 0.909 respectively. Results suggest the C spectrum is suitable for providing information required for remote sensing applications. Further research would

be focusing on implementing description in different azimuthal directions.

## 1 Introduction

The one-dimension wave spectra describe the energy distributions of the ocean waves over a certain range of wavenumbers or frequencies and are useful for simulation of the ocean surface roughness. The development of wave spectra dates back to 1952, when the Neumann spectrum was proposed based on theories and measurements in a simple form (Neumann, 1953), which

was applied by other follow-on spectra, such as Pierson Moskowitz (PM) spectrum (Pierson Jr. and Moskowitz, 1964), and





Joint North Sea Wave Observation Project (JONSWAP) spectrum (Hasselmann et al., 1973). Later in the 1990s, two of the spectra well applied today were proposed, one is the Goda spectrum modified from the JONSWAP spectrum by including the significant wave height and spectral peak period as variables (Goda, 1983). It was modified for describing the extreme waves at first and found later good in depicting the swells (Lucas and Guedes Soares, 2015), However, the spectral peak enhancement

factor $\gamma$ is taken as a constant with the value of 3.3, without further parameterization. Another is the Elfouhaily spectrum, which is an omnidirectional wave number range wind-wave spectrum using wind speed, inverse wave age, and wind fetch as the main variables. It has been well applied for research on electromagnetic wave scattering characteristics of rough sea surfaces (Elfouhaily et al., 1997). However, the value of the spectral peak enhancement factor $\gamma$ is also fixed, at 1.7.

The successful launch of the Ku-band radar in multiple incidence angles from nadir to 10°: the Surface Waves Investigation

and Monitoring instrument (SWIM) on board the China France Oceanography Satellite (CFOSAT) has achieved a better and continuous ocean wave observations on a large spatial scale since 2018 for wavelengths between 30 and 500m (Hauser et al., 2017). Before that, the Synthetic Aperture Radar (SAR) could also observe waves, but only for wavelengths larger than 200 m (Alpers et al., 1981). Moreover, altimeters can obtain ocean wave information by measuring the sea surface height and significant wave height from the sea wave echoes, but lack other accurate descriptions of the spectral details. The Infrared

Atmospheric Sounder Interferometer (IASI) has also been applied to obtain the probability density function of the wave slope in accordance with Cox and Munk's theory (Cox and Munk, 1954) and successfully obtained a series of measurements of the real sea surface. However, the interactions between the individual wave numbers are not specified, and are not globally available (Guérin et al., 2023).

When Comparing the Goda and Elfouhaily spectra with SWIM measurements in varied sea states, on one hand, the Goda

spectrum fits well with SWIM measurements for swell dominant sea state, but the "tail" for wavenumber larger than $k_p$ of the Goda curvature spectrum is lower than the SWIM measurements. The difference gradually increases with the increased wavenumbers. On the other, the Elfouhaily spectrum can represent the wind-wave well and the "tail" in its curvature spectrum demonstrates the same trend with the SWIM measurements (Wang et al., 2023).

Then in this research, based on the two widely applied spectra, a Combined spectrum (C spectrum) is derived for a better

description of the state-varied sea. The swell and wind wave terms are integrated while the sea-state is considered in a relevant variance—spectral peak enhancement factor $\gamma$, instead of the fixed values in the two spectra, and is parameterized from the SWIM observations. Validation of the C spectrum against the SWIM product not applied in the fitting process, and buoys data, indicating that the C spectrum model can provide a better description of the sea surface.





## 2 Method

### 2.1 The Goda and Elfouhaily spectrum against the measurements from SWIM

The Goda spectrum (G spectrum) is modified from the JONSWAP spectrum, by including the significant wave height and spectral peak period(or spectral peak wavenumber)as inputs, while the spectral peak enhancement factor $\gamma$ is set to 3.3. This led to a better representation of swells, but the wind speeds affecting the wind-waves, which in a mixing sea state entangles the swells are not included as a variable. The G spectrum is described by the following equation (Goda, 1983):

$$G(k) = \frac{1}{2}\alpha_* k^{-3} L_{PM} J_P \tag{1}$$

In Eq. (1), $k$ is wavenumber, $\alpha_*$ is the swell term (Goda, 1983):

$$\alpha_* = \frac{0.0624}{0.230 + 0.0336\gamma - 0.185(1.9+\gamma)^{-1}} H_{\frac{1}{3}}^2 k_p^2 \tag{2}$$

$\alpha_*$ is the function of spectral peak enhancement factor $\gamma$, the significant wave height $H_{1/3}$ and the spectral peak wavenumber $k_p$. It is the key for considering $\gamma$, which describes the peak sharpness. $L_{PM}$ in Eq. (1) is the Pierson-Moskowitz shape function, and is expressed as (Pierson Jr. and Moskowitz, 1964):

$$L_{PM} = \exp\left[-\frac{5}{4}\left(\frac{k_p}{k}\right)^2\right] \tag{3}$$

Where $k_p$ is the only variable.

And $J_P$ in Eq. (1) is a peak shape factor describing the inverted transfer of wave energy to larger scales from smaller scales caused by the nonlinear effects near the spectral peak, that was proposed by Hasselman in the JONSWAP spectrum (Hasselmann et al., 1973):

$$J_P = \gamma^{\exp\left[-\frac{\left(\sqrt{\frac{k_p}{k}}-1\right)^2}{2\sigma^2}\right]} \tag{4}$$

Here for G spectrum, $\gamma$ is set as a constant value: 3.3, and in Eq. (4) $\sigma$ depicts the spectral width in the vicinity of the spectral peak which is set as in Eq. (5).

$$\sigma = \begin{cases} \sigma_a = 0.07 & k \le kp \\ \sigma_b = 0.09 & k > kp \end{cases} \tag{5}$$

For G spectrum, $\sigma$ is set to 0.07 on the left side of the peak and 0.09 on the right side of the peak.

Elfouhaily spectrum (E spectrum) was developed based on *in-situ* measurements and experimental data and covers the full wavenumbers range. It introduces an analytical form suitable for modelling electromagnetic wave scattering over sea surfaces that aligns with Cox and Munk's theory. The omnidirectional curvature spectrum of long waves is specifically given by the following Eq. (6) (Elfouhaily et al., 1997):

$$S(k) = k^{-3}[B_l + B_h] \tag{6}$$





In Eq. (6), $B_l$ and $B_h$ are the low and high wavenumbers curvature spectrum respectively while the latter is not in the wavenumber range of SWIM measurements thus not included in this research. $B_l$ can be expressed as (Elfouhaily et al., 1997):

$$B_l = \frac{1}{2}\alpha_p \frac{c_p}{c} F_p \tag{7}$$

In Eq. (7), $\alpha_p$ is the Phillips and Kitaigorodskii equilibrium parameter for wind-waves, and is related with inverse wave age $\Omega$.

$$\alpha_p = 6 \times 10^{-3}\Omega \tag{8}$$

$\Omega$ is an important variable describing the sea states, and is affected directly by the wind speed at 10-meter-height ($u_{10}$).

$$\Omega = u_{10}\sqrt{\frac{k_p}{g}} \tag{9}$$

Where g is the Gravitational acceleration. For the ratio parameters in Eq. (7), $c_p$ is the phase velocity of the dominant wave, and $c$ is the phase velocity, and $\frac{c_p}{c}$ is an adjustment to the exponent position of $k$ proposed by Donelan and Pierson Spectrum (Donelan et al., 1985):

$$\frac{c_p}{c} = \frac{k}{k_p} \tag{10}$$

The ratio in Eq. (10) influences the increasing rate of the spectrum curve in wave numbers larger than the spectral peak wavenumber $k_p$, which is expressed as

$$k_p = \frac{2\pi}{\lambda_p} \tag{11}$$

From the dominant wavelength $\lambda_p$ obtained in the SWIM measurements, $k_p$ can be obtained.

For the last term in Eq. (7), $F_p$ is called the long wave side effect function (Elfouhaily et al., 1997).

$$F_p = L_{PM}J_p Lim \tag{12}$$

Where $L_{PM}$ and $J_p$ are the same as in Eq. (3) and Eq. (4), Here $\gamma$ in $J_p$ in the E spectrum is related to inverse wave age as follows (Elfouhaily et al., 1997):

$$\Upsilon = \begin{cases} 1.7 & 0.84 < \Omega < 1 \\ 1.7 + 6\log(\Omega) & 1 < \Omega < 5 \end{cases} \tag{13}$$

The $\sigma$ for $J_p$ in Eq. (4) for the E spectrum is also expressed as a function of the inverse wave age $\Omega$ (Elfouhaily et al., 1997):

$$\sigma = 0.08(1 + 4\Omega^{-3}) \tag{14}$$

And $Lim$ in Eq. (12) is a cut-off limiting the energy of the spectrum in wave numbers smaller than 10 times of $k_p$. It is related to the inverse wave age $\Omega$ , to explain the nonlinear interactions between different wave components (Elfouhaily et al.,





1997):

$$Lim = \exp\left\{-\frac{\Omega}{\sqrt{10}}\left[\sqrt{\frac{k}{k_p}} - 1\right]\right\}$$  (15)

Since in Eq. (15), *Lim* is varied with the inverse wave age $\Omega$, it varies with wind speeds $u_{10}$. The inclusion of this term stoops the growth rate of the E spectrum.

For the SWIM measurements, they provide the descriptions of real sea that is by nature of different states. Although it is focusing on longer waves in the spectrum, there is still a part of wind dominant range based on SWIM measurements and the corresponding sea state classification standard (Hauser et al., 2009; Xu et al., 2022; Hwang, 2009), the swell part (pure swell or swell dominant sea state) reaches up to 84.910% of all, while the pure wind wave is up to 7.952%, which can be classified fully developed, mature, and young wind wave according to the inverse wave age.

In different sea states, the comparison between the G spectrum and SWIM measurements reveals distinct characteristics. In the swell part, the G spectrum closely matches SWIM measurements in the range of 0.01 to 0.04 rad·m$^{-1}$ (157~628 m), with lower energy between 0.04 to 0.2 rad·m$^{-1}$ (31~157 m) (Wang et al., 2023). For the pure wind wave range, in the fully developed sea conditions, the G spectrum exhibits higher energy in the range of 0.03 to 0.06 rad·m$^{-1}$ (104~209 m), with lower energy between 0.06 to 0.2 rad·m$^{-1}$ (31~105 m). In mature seas, the G spectrum shows higher energy between 0.05 to 0.1 rad·m$^{-1}$

(63~126 m), with lower energy between 0.1 to 0.2 rad·m$^{-1}$ (31~63 m). At the same time, the E spectrum considers the inverse wave age in the spectral peak shape function $J_p$ in Eq. (4), thus is effective for waves with the inverse wave age $\Omega$ greater than 0.84 (Elfouhaily et al., 1997). In comparison with the SWIM measurements, E spectrum exhibits higher energy in the range of 0.01~0.2 rad·m$^{-1}$ (31~628 m) in fully developed wind wave and mature wind-wave, while in swell parts for the lower energy in the range of 0.01~0.12 rad·m$^{-1}$ (52~628 m), and it aligns closely with SWIM measurements in the range of 0.12~0.2 rad·m$^{-1}$

$^{-1}$ (31~52 m) (Wang et al., 2023). In all, the SWIM measured spectral width aligns more closely with the G spectrum, while the E spectrum exhibits greater variability in spectral width suggests modifications in the E spectrum on its variance of inverse wave age can fit SWIM observations well. Therefore, in this research, a spectrum model simulating SWIM measurements of the natural sea surface is established by combining the corresponding terms in G and E spectra. By inheriting the spectral width factor from the G spectrum as one of the shape parameters, while simulating the mixed sea states induced wind energy by the

lower wavenumber part of the E spectrum, the C spectrum model is hereafter established.

**2.2 The C spectrum**

In the mixed sea state, the swell and wind-wave discriminating wave number can be vague, while the real sea surface integrates the features of both swell and wind-wave spectra. Following the previous section, a C spectrum is established from the G and E spectra to simulate SWIM measurements. The form of the G spectrum is applied as the basic expression of the C spectrum

following the measuring wavenumber range of SWIM. Following the right side of the spectral peak wavenumber $k_p$ of the G spectrum, the term $\frac{c_p}{c}$ in Eq. (7) of the E spectrum is introduced to adjust the slope of spectra to fit the SWIM measurements, and the term *Lim* in Eq. (15) with inverse wave age $\Omega$ simulate the relation of sea state variance due to wind speed modelled





in $u_{10}$ in the lower wavenumbers, and are included in the C spectrum model. Effectively on the shape of the spectrum, the inclusion of this term avoids the spectrum "tail" being too large thus to be diverted from the measurements. Then specifically, the C spectrum is modelled as:

$$S(k) = 2\pi^2 \alpha k^{-3} L_{PM} J_P \frac{c_p}{c} \exp\left\{-\frac{\Omega}{\sqrt{10}}\left[\sqrt{\frac{k_p}{k}} - 1\right]\right\} \tag{16}$$

$$\alpha = \frac{0.0624}{0.230 + 0.0336\gamma - 0.185(1.9 + \gamma)^{-1}} \delta^2 \tag{17}$$

$$\delta = \frac{H_{\frac{1}{3}}}{\lambda_p} = \frac{H_{\frac{1}{3}} k_p}{2\pi} \tag{18}$$

In Eq. (16), $\alpha$ is inherited from the G spectrum in the form considering the wave steepness $\delta$ expressed in Eq. (18), related with the significant wave height and spectral peak wavenumber. $\alpha$ is important for swell components, and $\gamma$ in $\alpha$ affects the shape of the spectrum peak. Moreover, in Eq. (16), $k^{-3}L_{PM}J_P$ is the same as in the Goda spectrum. As explained in the former content, the inclusions of the last two items from the E spectrum adjust the growth rate of the curve in the spectral tail. Furthermore, in Eq. (17), $\gamma$ also reflects the sharpness of the spectral peak and is an indication of the concentration level of wave energy, and is generally used as an empirical parameter. As noted in previous content, in the Goda and the JONSWAP spectra, the parameter is set to 3.3(Hasselmann et al., 1973). In 1977~1980, $\gamma$ was related to the dimensionless wind fetch based on the nearshore observations (Mitsuyasu, 1977; Mitsuyasu et al., 1980). In 1985, $\gamma$ was further related to the wave development in addition to the dimensionless wind fetch based on limited buoys observation (Karaev et al., 2008). When the wind fetch can also be related to the inverse wave age in the E spectrum (Elfouhaily et al., 1997). At the same time, the wave steepness $\delta$ is related with significant wave height and is important in characterizing the swell. Following that development, in this research, the inverse wave age $\Omega$ and wave steepness $\delta$ are applied in modelling $\gamma$. At the same time, to fit the SWIM observations, the objective condition is set as the spectral peak from the C spectrum equals the measured peak value:

$$S_{max} = S(k_p) \tag{19}$$

Then the spectral value corresponding to the spectral peak wavenumber can be calculated according to the C spectrum value at $k_p$:

$$S(k_p) = \frac{1}{2}\frac{5\gamma}{\left(1.15 + 0.1688\gamma - \frac{0.925}{1.909 + \gamma}\right)}\frac{Hs_{\frac{1}{3}}^2}{16 k_p} e^{-1.25} \tag{20}$$

From Eq. (19) and Eq. (20), providing the significant wave height $H_{s\frac{1}{3}}$, $k_p$ and $S(k_p)$, the spectral enhancement factor $\gamma$ can be obtained. $S(k_p)$ is the corresponding value of the C spectrum model at spectral peak wavenumber $k_p$. And according to the objective expressed in Eq. (19), $S_{max}$ is the maximum value of the measured data.



## 2.3 The evaluation indices

In this research, the C spectrum is evaluated from the Difference index (DI) and the R-squared ($R^2$), expressed in Eq. (21) and (23) respectively. In Eq. (21), $S(k_i)$ represents the spectral model value, and the $\hat{S}(k_i)$ corresponds to the SWIM measurements, and $m0$ denotes the $0^{th}$-order momentum of the wave spectrum given by Eq. (22). The DI offers a comprehensive evaluation of the distinction between the spectral models and SWIM measurements in total energy, and a lower DI value indicates a smaller discrepancy between the model and measurements, thus a more precise fit.

$$DI = \frac{1}{m_0} \int_{k_{min}}^{k_{max}} |S(k_i) - \hat{S}(k_i)| dk \qquad (21)$$

$$m_0 = \int_{k_{min}}^{k_{max}} \hat{S}(k) dk \qquad (22)$$

$R^2$ in Eq. (23) is calculated by computing the ratio of 1 minus the sum of squared residuals to the total sum of squares between $S(k_i)$ and $\hat{S}(k_i)$, which represents model value and SWIM measurements, respectively, and $\bar{S}(k)$ means the average of the model. The $R^2$ approaching 1 indicates that a smaller sum of squared residuals, and the model value are closer to the SWIM

measurements.

$$R^2 = 1 - \frac{\sum_{i=1}^{n} \left(S(k_i) - \hat{S}(k_i)\right)^2}{\sum_{i=1}^{n} \left(S(k_i) - \bar{S}(k)\right)^2} \qquad (23)$$

## 3 Experiments and Results

### 3.1 Data Descriptions

#### 3.1.1 The SWIM wave product

The in-situ observation in this research is provided by the Surface Waves Investigation and Monitoring (SWIM) carried out by the China France Oceanography Satellite (CFOSAT). SWIM products provide valuable wave spectrum details as well as wave parameter information in global coverage. The L2 spectra product is used in this research for fitting the $\gamma$, while the set not applied in fitting is for evaluation of the C spectrum. The SWIM L2 wave product provides of the measured omnidirectional spectrum, 10-meter-height wind speed from European Centre for Medium-Range Weather Forecasts (ECMWF), and wave

parameters including significant wave height and dominant wavelength. Among them, the peak wavenumber $k_p$ is derived from the dominant wavelength while from the wind speed and $k_p$, the inverse wave age can be obtained from the products. The SWIM L2 10 ° beam observations have been validated the best consistency with the mode spectrum and buoy observations (Hauser et al., 2021) and are applied for this research.





Specifically, the 2021 CFOSAT-SWIM L2 10° global products are used for C spectrum modelling and the 2022 products are used for validation. First, the land, rainfall flags, and confidence interval variables are utilized to filter out pure sea surface and rain-free wave cells (referred to as "box", generally a 70×90 km sized square on the ocean surface). Subsequently, to suppress abnormally high values at low wave numbers of SWIM caused by parasitic peaks (Merle et al., 2021; Xu et al., 2022), a criterion is set where the difference between the spectral peak wave number and the wave number corresponding to the spectral maximum value does not exceed 0.007. Then, according to the histogram of $\gamma$, the boxes for $\gamma$ ranging from 1 to 12 are

selected as the research data that depicts the sea surface well. In total, 1 251 067 boxes are found as the experiment data in 2021 for modelling, and 1 307 256 boxes in 2022 are obtained for validation.

### 3.1.2 NDBC Buoys

In another validation process, the buoy observations from the National Data Buoy Center (NDBC) are applied as the *in situ* measurements in addition to the SWIM products. The buoy product used in this research is spectral wave density measured

offshore 100 km in 2022, with 47 frequencies ranging from 0.020 to 0.485 Hz (NDBC Web Data Guide, 2023). The SWIM data varies with the wavenumber ranging from $0.01 \sim 0.2$ rad·m$^{-1}$, and the buoy frequency spectra is transferred into wavenumber spectra and the linear interpolation is applied to them to unify the unit and scale. The time lag between buoy measurements and the SWIM product applied in the validation is 1 h, and the space interval is 50 km. In all, 1701 match-pairs of buoys and SWIM are obtained from 37 buoy stations. As shown in Fig. 1, the yellow diamond marks are the distributions of those buoy

stations. They are mainly distributed between the latitudes of 20 to 60 ° N, and the longitudes ranging from 50 ° W to 160 ° E.

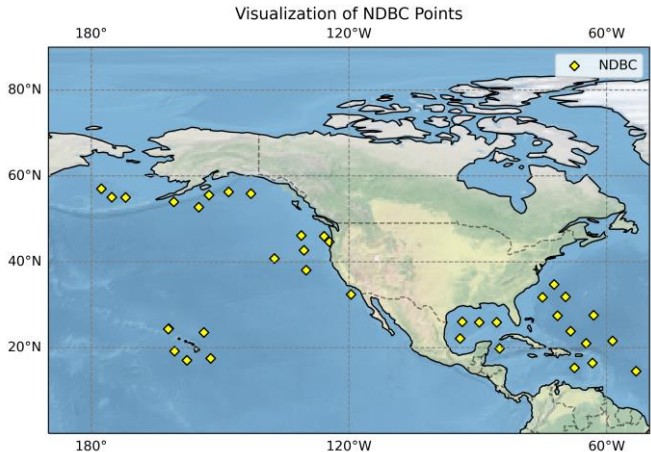

**Figure 1: The distribution of selected buoy stations**



### 3.2 The retrieval of parameter $\gamma$

To establish the function of $\gamma$ from $\delta$ and $\Omega$, the data is divided into multiple grids. Since the curve shapes of $\gamma$ against $\Omega$ for each $\delta$ closely resembles trigonometric functions, the Fourier series is chosen to fit $\gamma$ with $\Omega$ in each $\delta$ interval. Considering the range of $\delta$ observed by SWIM for the optimal fitting order, the SWIM boxes are divided into two parts according to $\delta$. For $\delta$ ranges from 0.004 to 0.0115, the $R^2$ of the second-order Fourier function fitting for $\gamma$ with the inverse wave age $\Omega$ are almost all greater than 0.8. For the remaining of $\delta$ range between 0.0115 and 0.0295, the most $R^2$ from the third-order Fourier

function fitting for $\gamma$ with the inverse wave age $\Omega$ is above 0.85.

(1) The fitting results when $\delta$ is between 0.004~0.0115
The fitting results are as shown in Eq. (22):

$$\gamma = \frac{a_0}{2} + \sum_{n=1}^{1} a_n \cos n\pi\Omega \tag{22}$$

The coefficients $a_0$,$a_1$ are fitted with wave steepness based on the appropriate functional form, and the relationship between

225 the coefficients and the wave steepness are as follows:

$$a_0 = 0.003 * \delta^{-1.156} \tag{23}$$

$$a_1 = 1.274 \ln \delta + 9.536 \tag{24}$$

Fig.2 depicts the fitting curves of $a_0$, and $a_1$ with the wave steepness $\delta$. The relationship between $a_0$ and the wave steepness follows a power function, with a goodness of fit of 0.830. The relationship between $a_1$ and the wave steepness $\delta$ follows a

230 logarithmic function, with a goodness of fit of 0.885.

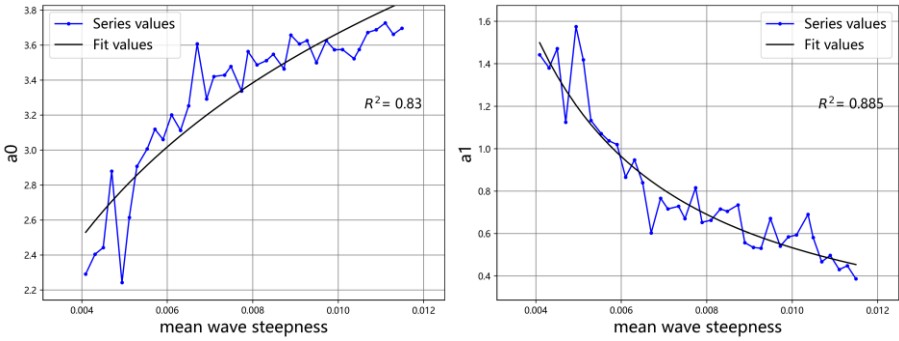

**Figure 2: The fitness of $a_0$, $a_1$ with the wave steepness**

(2) The fitting results when $\delta$ is between0.0115~0.0295

For the wave steepness range of 0.015 to 0.0295, inverse wave age and $\gamma$ were fitted using a third-order Fourier series. The

235 final fitting results are as shown in Eq. (25):

$$\gamma = \frac{a_0}{2} + \sum_{n=1}^{2} a_n \cos n\pi\Omega \tag{25}$$





The fitting relationships between $a_0$, $a_1$, $a_2$, and wave steepness are as follows:

$$a_0 = 3.132 e^{(12.273\delta)} \tag{26}$$

$$a_1 = -0.365 \ln\delta - 1.21 \tag{27}$$

$$a_2 = -0.093 \ln\delta - 0.442 \tag{28}$$

In Fig 3, the fitting curves of $a_0$, $a_1$, $a_2$, and wave steepness are displayed. The relationship between $a_0$ and wave steepness follows an exponential function, with a high $R^2$ of up to 0.949. The relationship between $a_1$ and wave steepness conforms to a logarithmic relationship, with a $R^2$ as high as 0.918. For $a_2$ and wave steepness, a logarithmic fitting approach is adopted, resulting in an $R^2$ of 0.635.

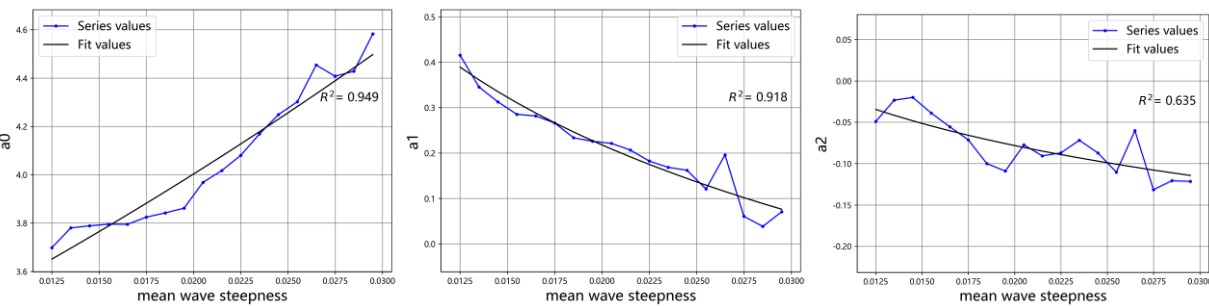

**Figure 3: The fitness of $a0$ , $a1$ , $a2$ with wave steepness**

### 3.3 The established C spectrum

The C spectrum takes the spectral peak wavenumber $k_p$, the 10-meiter-height wind speed $u_{10}$, and the significant wave height $H_{\frac{1}{3}}$ as variables. $k_p$ is the derived from the SWIM measurements. The inverse wave age $\Omega$ can be then be calculated from kp

and $u_{10}$, and the wave steepness $\delta$ can be obtained with $k_p$ and $H_{\frac{1}{3}}$. Fig. 4-5 depict the curve change of the C spectrum with different three input variable combinations.

Fig. 4 (a)~(c) depict respectively the C spectrum in height spectrum, slope spectrum, and curvature spectrum at a spectral peak wavenumber $k_p$= 0.048 rad·m$^{-1}$ and a wind speed $u_{10}$ of 10 m·s$^{-1}$ under different significant wave heights $H_{\frac{1}{3}}$. In Fig. 4, the $k_p$ and $u_{10}$ are fixed for the $H_{\frac{1}{3}}$ is variance from the spectrum. $\delta$ ranges from 0.008 to 0.069, with the wave age $\Omega$ fixed at

255 0.699. With increasing $H_{\frac{1}{3}}$, the $\delta$ also increases, and the area under the spectral lines increases. In (a) and (b), the trend of change is almost identical to the G spectrum, with the spectral peak in the C spectrum slightly smaller than that in the G spectrum. In (c), the values of the C spectrum on the right side of the spectral peak gradually increase, contrasting with the trend in the G spectrum where the curve gradually flattens. The curve is more sensitive to changes in $H_{\frac{1}{3}}$ than to variations in $u_{10}$ and $k_p$.





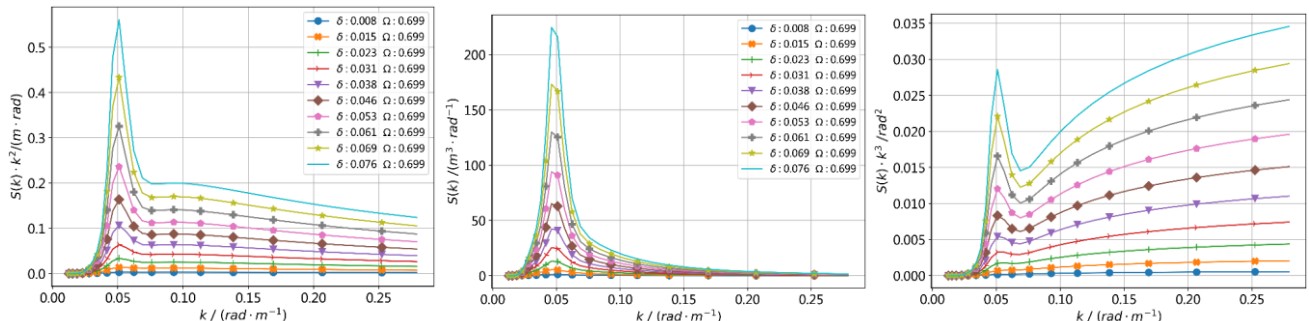

**Figure 4: The C spectrum at different $H_{\frac{1}{3}}$, when $k_p$ is 0.048 rad·m⁻¹ and $u_{10}$ is 10 m·s⁻¹, (a) is Combined height spectrum, (b) is Combined slope, (c) is Combined curvature. δ is wave steepness, and Ω is inverse wave age.**

Fig. 5 (a)~(c) also illustrate the C spectrum in height spectrum, slope spectrum, and curvature spectrum, but for different $k_p$ at $H_{\frac{1}{3}}$ of 3 m and $u_{10}$ of 10 m·s⁻¹. Since the $k_p$ varies, both the δ and the Ω change accordingly, with their ranges being 0.016 to 0.034 for the δ and 0.319 to 1.463 for the Ω from this setting. As the $k_p$ increases, the δ increases, and the Ω decreases. And as the $k_p$ increases, the spectral peak shifts to the right. In (b), as the $k_p$ increases, the area under the curve gradually decreases, indicating a transition from swell-dominated to wind wave-dominated sea states.

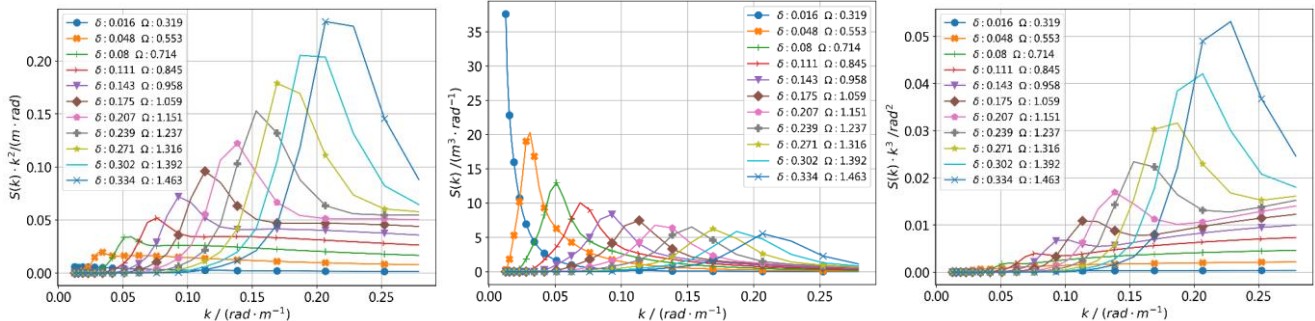

**Figure 5: The C spectrum at different $k_p$, when $H_{\frac{1}{3}}$ is 3 m and $u_{10}$ is 10 m·s⁻¹, (a) is Combined height spectrum, (b) is Combined slope, (c) is Combined curvature**

Fig. 6 (a)~(c) are also the C spectrum in height, slope, and curvature spectrum, now for different $u_{10}$ at the $H_{\frac{1}{3}}$ of 3 m and the $k_p$ of 0.048 rad·m⁻¹. When now $H_{\frac{1}{3}}$ and $k_p$ are fixed, the δ is a fixed value, as in the legend, while the Ω varies. As the $u_{10}$ increases, the Ω gradually increases. The parts affected by $u_{10}$ are mainly located to the right of the $k_p$ and at the spectral peak. The variation to the right of the $k_p$ is due to the addition of the Lim term, while the variation of the spectral peak is due to the fitting of γ and sea state variables. In particular, the change in the right side of the height spectrum due to $u_{10}$ is relatively small. However, in the slope spectrum and curvature spectrum, it can be clearly observed that as the $u_{10}$ increases, indicating an increase in the Ω, the wave energy reflected by the area under the curve gradually decreases, indicating a transition of sea state towards $u_{10}$.





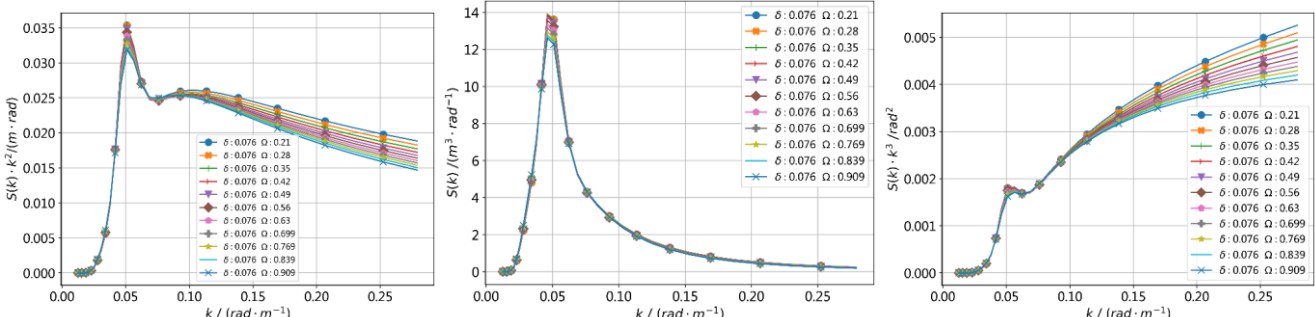

**Figure 6: C spectrum at different wind speed, when significant wave height is 3m and $k_\mathrm{p}$ is 0.048 rad·m$^{-1}$, (a) is Combined height spectrum, (b) is Combined slope, (c) is Combined curvature**

### 3.4 Evaluation of the established C spectrum

In this section, the evaluation by the DI and $R^2$ described in Eq. (21) and (23) are achieved. For the wave steepness $\delta$ range of evaluation results is 0.0125 to 0.0295 (corresponding to the third-order Fourier expansion of the spectral peak enhancement factor $\gamma$ in the C spectrum) are provide alone since the evaluation results of the second-order fitting models in the other range of $\delta$ are consistent with this part and is not shown. First, 2022 SWIM L2 measurements not involved in fitting $\gamma$ is used while for validation involving buoy measurements, the 2022 NDBC measurements are applied. The DI and $R^2$ of the C, G and E spectra are calculated in the form of height and curvature.

### 3.4.1 The results of evaluation

**A. SWIM measurements as *in situ* measurements**

The two-dimensional histogram is applied to visualize the results of the two indices by comprising the C spectrum to the G spectrum as in Fig. 7, wherein (a) and (b), the x-axes are DI and $R^2$ of G height spectrum, and the y-axes are DI and $R^2$ of the C height spectrum. As shown in (c) and (d), the comparison in DI and $R^2$ of the C and the G curvature spectra. The blue line is generated by setting $y = x$, and the red line is a linear fitting of the data.

From Fig. 7 (a) and (c), DI of the most data points lie below the y < x, and the dominant proportion is shown in Table 1. The data of C spectrum primarily cluster in the yellow area, where the DI are less than 0.3. Fig. 7 (b) and (d) are comparisons of $R^2$, and the most $R^2$ in the C spectrum are higher than those in the G spectrum. Moreover, the $R^2$ of the C spectrum is predominantly distributed above 0.8. It is evident that the C spectrum is better than those of the G spectrum in fitting SWIM measurements





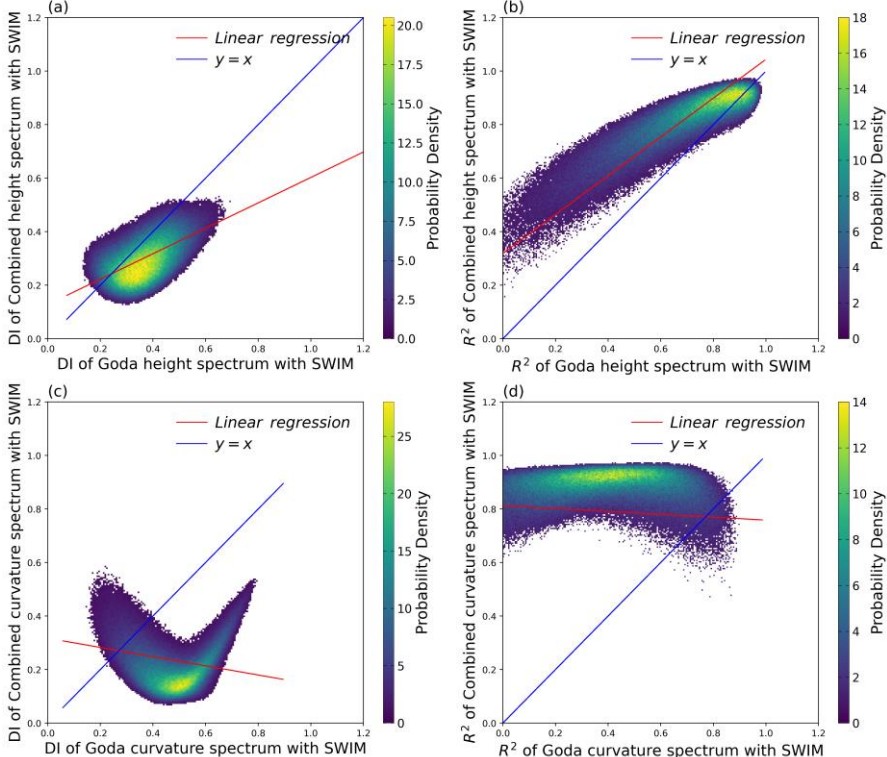

**Figure 7: This is the comparisons of the validation index about G spectrum with C spectrum with SWIM in 2022, (a), (c) is DI of height and curvature spectrum, (b), (d) is $R^2$ of height and curvature spectrum.**

Fig. 8 is the comparison of the E spectrum with the C spectrum, which has the same setting with Fig. 7. According to the two DI comparison results on the left as shown in Fig. 8, both the C height and curvature spectra exhibit absolute advantages in DI, and data primarily concentrated in the y < 0.4 interval. As for the two $R^2$ comparison results on the right, the C height and curvature spectra show dominance ratios shown in Table 1 with data mainly concentrated in the y > 0.7 interval. This indicates that compared to the E spectrum, the C spectrum shows better conformity with SWIM-measured data.



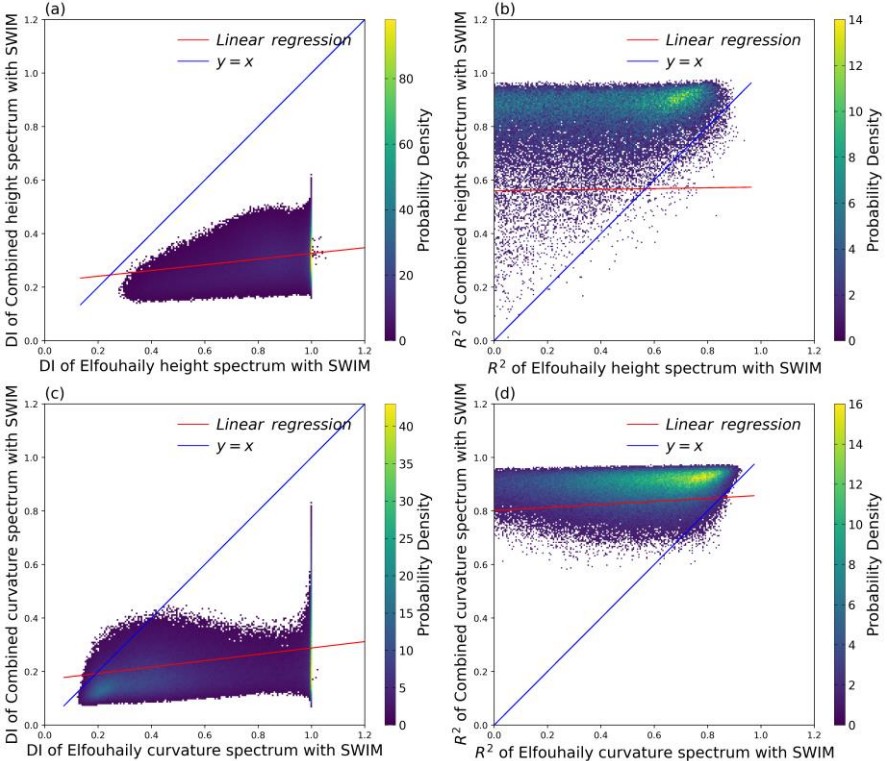

**Figure 8: This is the comparisons of the validation index of E spectrum with C spectrum, (a), (c), (e)is DI of height , curvature and slope spectrum, (b), (d),(f) is R2 of height, curvature, and slope spectrum.**

The value of the two indices of two spectra against SWIM measurements are specified below the Table 1. The DI approaching 0 indicates the better fitting. The proportion of curvature and height spectrum in DI about the C spectrum better than the G spectrum is 82.9% and 78.0%, which means the C spectrum fits more the SWIM measurements. The $R^2$ of the C height curvature spectra exceed the G spectrum by 99.3%, and 91.8%. The DI and $R^2$ proportion of the C spectrum better than the E spectrum in fitting SWIM measurements are above 90%. Then conclusion can be drawn that the C spectrum has better indexes with SWIM measurements than the G spectrum and E spectrum respectively, thus baring features of the ocean surface closer to the real scenes for remote sensing observations.

Table. 1. This is the comparison of the three spectra in fitting SWIM measurements

| | The C spectrum better than the G spectrum | The C spectrum better than the E spectrum |
|---|---|---|
| DI of curvature | 0.829 | 0.918 |
| DI of height | 0.780 | 0.993 |
| $R^2$ of curvature | 0.851 | 0.962 |



| Table. 1. This is the comparison of the three spectra in fitting SWIM measurements (continued) | | |
|---|---|---|
| $R^2$ of height | 0.909 | 0.983 |

**B. NDBC buoys measurements as *in situ* measurements**

To verify the performance of the C spectrum compared with G and E spectra on the real sea state, the 2022 NDBC buoys wave spectra data after spatial-temporal matching is as *in-situ* measurements in the second evaluation reference. The criteria for
selecting data and the legend of Fig 10-11 are consistent with the 3.3.1 about SWIM as *in-situ* measurements Sect.

Fig. 9 (a), most DI of the C height spectrum is lower than those of the G height spectrum, but the proportion of superiority is not as significant, and the C spectrum has a superiority proportion with DI concentrated in the region where y < 0.5. In Fig. 9 (b), the $R^2$ for the C height spectrum is superior and mainly concentrates in the region where y > 0.8. In Fig. 9 (c) and (d), the DI and $R^2$ of the C curvature spectrum still have a superiority. The data are concentrated near x = 0.5, y < 0.4, and near x =
0.6, y > 0.8. Results indicate the C spectrum has a good fit with NDBC buoy measurements than the C spectrum. The superiority proportions in DI and $R^2$ for the C spectrum are shown in Table 2.

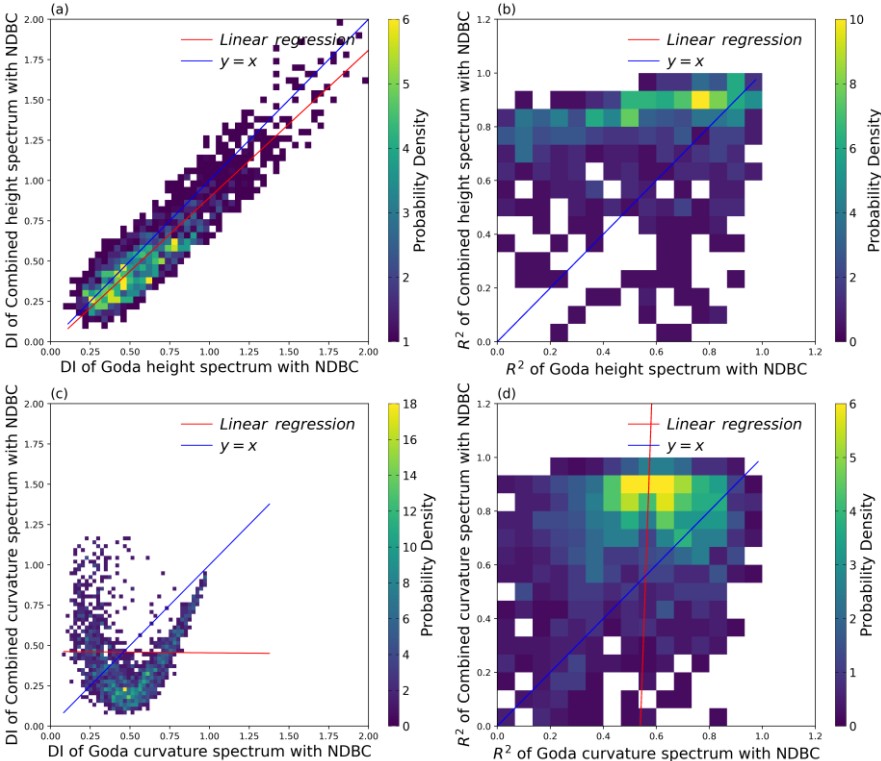

**Figure 9: This is the Comparisons of the validation index of G spectrum and C spectrum with buoys, (a), (c) is DI of height and curvature spectrum, (b), (d) is R² of height and curvature spectrum.**





Fig. 10(a) compares the DI between the C and E height spectra. The superiority ratio of C height spectrum is 0.830, and the DI is mainly concentrated around x=1 and y<0.5. Fig. 10 (b) displays the $R^2$ of the C height spectrum with a superiority proportion. The data are mainly concentrated in the region where y>0.8. Fig. 10 (c) and (d) depict the comparison between the C and the E curvature spectrum in terms of DI and $R^2$. The C curvature spectrum has a superior proportion than the E curvature spectrum, with data primarily concentrated in the intervals y < 0.4 and y > 0.8. The concentrated regions on the x-axis

correspond to poorer indicators. Therefore, the C spectrum is closer to NDBC data compared to the E spectrum and exhibits better conformity. The superiority proportions of DI and $R^2$ for the C spectrum compared with the E spectrum are shown in Table 2.

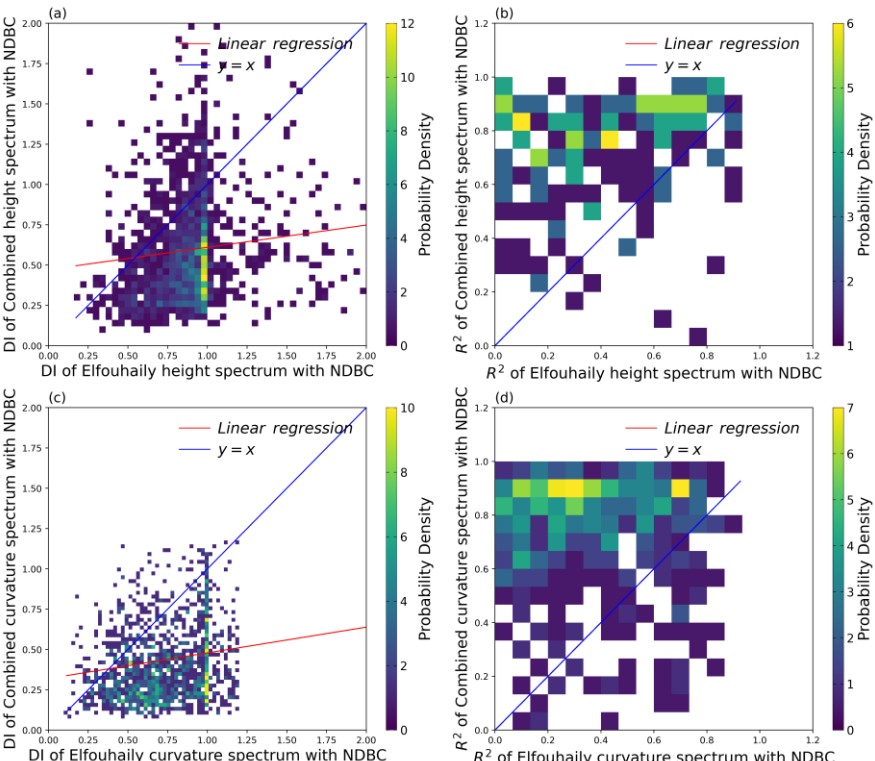

**Figure 10: This is the comparisons of the validation index of E spectrum and C spectrum with buoys, (a), (c) is DI of height and**
345 **curvature spectrum, (b), (d) is $R^2$ of height and curvature spectrum.**

Table 2 illustrates the better performance ratio of the C spectrum compare with G and E spectra. Compared to the G spectrum, the C height spectrum has a superior proportion of 71.9% in DI, while the C curvature spectrum has a superior proportion of 65.6%. The proportion of $R^2$ superiority for the C height spectrum is 85.6%, while for the C curvature spectrum, it is 63.6%. Compared to the E spectrum, the proportions where the C spectrum's DI and $R^2$ are superior is above 80%. The comparison

of three spectra demonstrates the advantages of the C spectrum in characterizing the measured data, aligning more closely with the NDBC measured data, further illustrating that the C spectrum provides a better representation of real sea states compared to the G and E spectra.





Table. 2 This is the comparison of indexes about different spectra with buoys

|  | C spectrum better than G spectrum | C spectrum better than E spectrum |
|---|---|---|
| DI of curvature | 0.656 | 0.855 |
| DI of height | 0.719 | 0.830 |
| $R^2$ of curvature | 0.636 | 0.831 |
| $R^2$ of height | 0.856 | 0.907 |

### 3.4.2 The performance in selected cases

One group of the swell and wind-wave boxes is randomly selected to investigate the representation effects under different sea states of three theoretical models: The C spectrum, the E spectrum, the G spectrum, and two measured data sets: SWIM and NDBC wave spectrum. The wave steepness of swell is 0.020, corresponding to an inverse wave age of 0.479, respectively. The wave steepness for the wind-wave was 0.026, with an inverse wave age of 0.803.

**Table. 3 The wave steepness, inverse wave age and indices of boxes**

| Wave steepness | Inverse wave age | DI | $R^2$ |
|---|---|---|---|
| 0.02 | 0.479 | 0.169 | 0.94 |
| 0.026 | 0.803 | 0.119 | 0.978 |

In Fig. 11, the C spectrum shows a good fit with the measured data, especially at the spectral peak and in the spectral tail data with wave numbers greater than 0.05 rad·m$^{-1}$. However, there is a slight discrepancy between 0.03 and 0.06 rad·m$^{-1}$, where the C spectrum is slightly larger than the NDBC data but slightly smaller than the SWIM data. The G spectrum consistently exhibits peaks greater than the two sets of measured data. Additionally, the G spectrum curves are slightly smaller than the measured data for wave numbers greater than 0.03 rad·m$^{-1}$. The E spectrum hardly represents the measured data at low wave numbers, but it closely matches the measured data for wave numbers greater than 0.09 rad·m$^{-1}$.

The left panel of Fig. 11 shows an enlarged view of the spectral peaks. The peak of the G height spectrum is close to 120, while the peaks for the SWIM and NDBC measured data are less than 100 and 80, respectively. After γ fitting, the C spectrum in the form of height spectrum is smaller than 90, and the error between the peak of the C spectrum and the NDBC spectrum is approximately 5%, while the error with the SWIM spectrum is about 10%. The error between the peak of the G spectrum and the NDBC spectrum is about 35%, and approximately 20% with the SWIM spectrum.





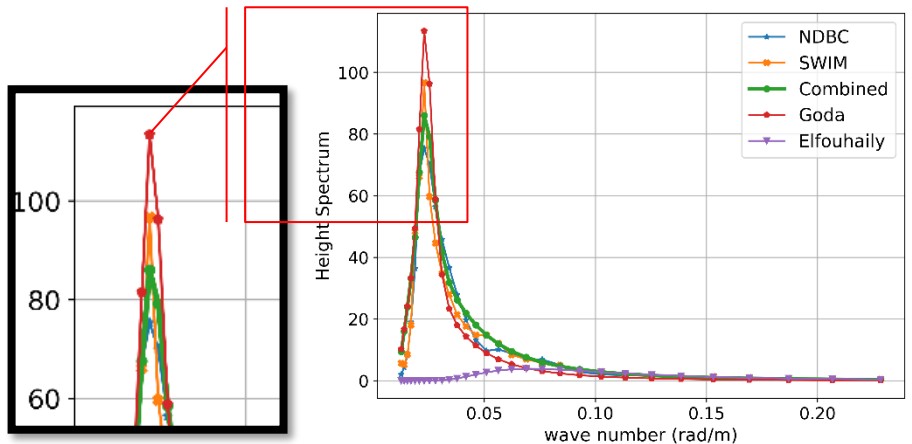

**Figure 11: Comparison of the SWIM, G, E and Combined height spectra when wave steepness is 0.02, and inverse wave age is 0.479.**

In Fig. 12, the SWIM and NDBC measured curvature spectra exhibit differences after 0.09 rad·m$^{-1}$, with SWIM measurements slightly larger than NDBC measurements. The advantage of the C spectrum becomes more evident towards the tail end, showing a trend more consistent with the measured data and achieving better numerical fit. Conversely, the trend of the G spectrum's tail end grows slowly with smaller numerical values, leading to an increasing gap with the measured data. After 0.09 rad·m$^{-1}$, the values and trend of the E spectrum are nearly identical to the measured data, especially aligning closely with

the SWIM data. However, the E spectrum hardly represents the measured data at low wave numbers.

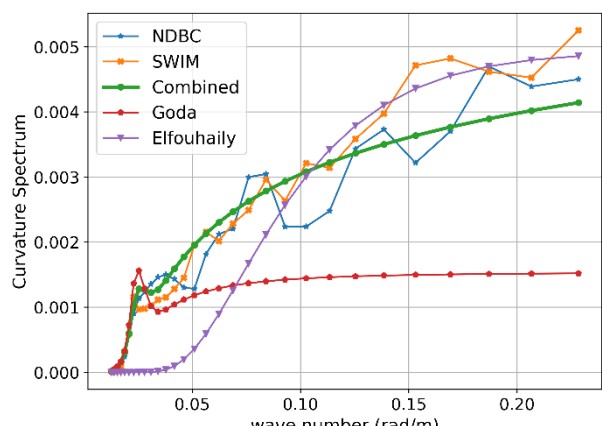

**Figure 12: Comparison of the SWIM, G, E and Combined curvature spectra when wave steepness is 0.02, and inverse wave age is 0.479.**

   In Fig. 13, the C spectrum shows good agreement with the measured data, especially in the peak and tail-end data with wave

numbers greater than 0.12 rad·m$^{-1}$, but there is a slight overestimation of the data between 0.07 and 0.12 rad·m$^{-1}$. The G spectrum underestimates the SWIM measured data slightly after 0.09 rad·m$^{-1}$, and the data is lower than the NDBC measured data after





0.06 rad·m⁻¹. The peak position and magnitude of the E spectrum differ significantly from the measured data, with the spectrum width wider than the measured data. The right end of the spectrum curve exceeds the measured data, indicating a poor representation of the wind-wave sea state by the E spectrum.

On the left panel of Fig. 13 is an enlarged view of the spectrum peaks. The peak of the G spectrum is close to 17.5 exceeding the measured data, whereas the measured peak is 12.5. After γ correction, the peak of the C spectrum is 12.5, and the error between the C spectrum and the two measured spectra peaks is negligible, almost zero. However, the error between the G spectrum and the two measured spectrum peaks is approximately 50%.

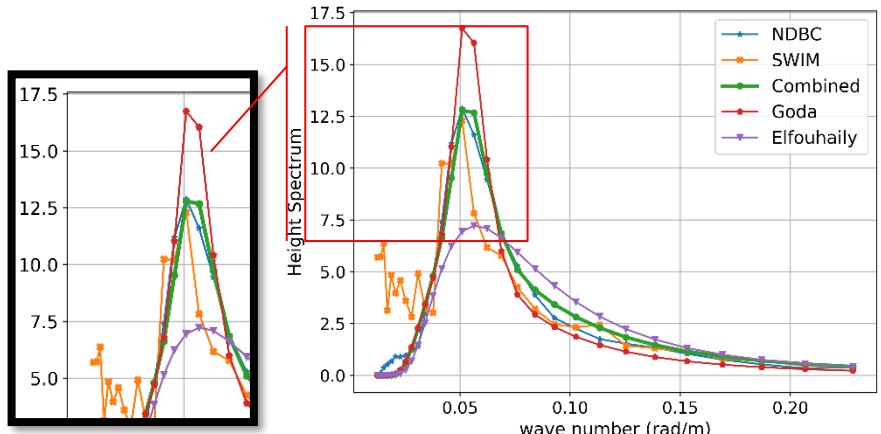

**Figure 13: Comparison of the SWIM, G, E and Combined height spectra when wave steepness is 0.026, and inverse wave age is 0.803.**

In Fig.14, the SWIM and NDBC measured curvature spectra exhibit differences after 0.05 rad·m⁻¹, but the overall trends of the two measured datasets are basically consistent. The C spectrum follows a consistent trend with the measured data, with values closely matching. The G spectrum shows slower growth in the tail end, with smaller values, leading to an increasing gap

between the model and the measurements. The E spectrum continuously rises overall, especially at the right end, almost above all other curves. In the curvature spectrum, its peak is hardly observed.



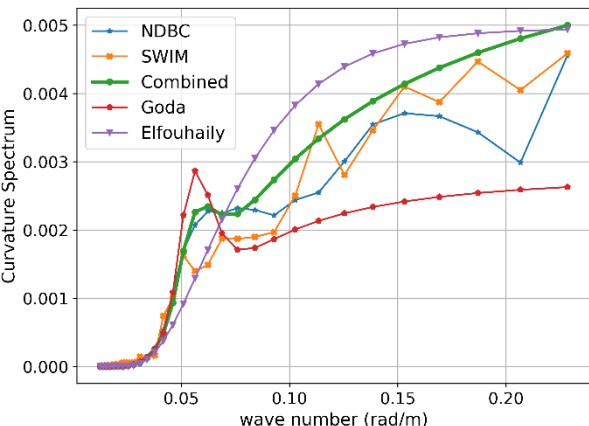

**Figure 14: Comparison of the SWIM, G, E and Combined curvature spectra when wave steepness is 0.026, and inverse wave age is 0.803.**

## 4 Conclusions

Most ocean wave spectra are established from limited *in-situ* observations and thus lack extensive support from large-scale measured information. For ocean wave spectra, the peak enhancement factor $\gamma$ is often taken as the fixed values in wave spectra. With the successful launch of the CFOSAT-SWIM, the larger scales of wave spectra are globally and continuously available. In this research, based on CFOSAT-SWIM L2 data, combined with the structural characteristics and performance of two spectra against SWIM-measured real sea states, the swell structure of the G spectrum and the wind-wave structure of the E spectrum are integrated with a C spectrum established. Then the peak enhancement factor $\gamma$ is fitted by sea state-related variables in two different classes divided by wave steepness for the C spectrum in the SWIM measurements. The validation of the C spectrum is achieved from SWIM together with NDBC buoys. The DI index and $R^2$ of the C spectrum, the G spectrum, and the E spectrum are calculated and compared. Then the indexes of the C spectrum with those of the G and E spectra are compared to determine the proportion of cases where the C spectrum is superior.

The conclusion of this research indicates that in the C spectrum, $\gamma$ influenced by the sea state is modelled, primarily affected by the combined effect of wave steepness and inverse wave age obtained from the significant wave height, wind speed at 10-meter height, and the spectral peak wavenumber (or dominant wavelength). An expression of the $\gamma$ with the wave steepness and inverse wave age jointly has been established for multiple sea states, and the nonlinear interaction between wind waves and swells can be confirmed again. The evaluation indexes of the C spectrum model show the highest proportion of superiority, indicating that the C spectrum is closer to the measured data compared to the G and E spectra. This also demonstrates the C spectrum's ability to represent real sea states to a certain extent bounded by the measuring range of SWIM product applied in the model establishment, where the C spectrum reflects well the energy distribution in wave numbers under measured sea states, thus can provide ocean surface model for related research and applications.



The directional variance in this research is not discussed yet, and is our further research based on the established C spectrum.

**Author contributions**

Yihui Wang designed the experiments including programming, data analysis, modelling, and validation, while finished the original draft.

Xingou Xu proposed the research idea, responded to the oversight and leadership for the research activity planning and
execution, reviewed and modified the draft.

**Competing interests**

The authors declare that they have no conflict of interest.

**Acknowledgments**

This research is partly supported by the task "Research on aerospace remote sensing techniques and algorithms for the key
oceanic and meteorological parameters" of the National key R&D project under the grant number "2022YFE0204600", and
partly supp by the project 2023CFO010 supported by Key Laboratory of Space Ocean Remote Sensing and Application, MNR.

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
