# Peer review of "An Improved One-dimensional Ocean Wave Description based on SWIM Observations"

_Atmospheric Measurement Techniques, 2024_

## Author Comment (AC2)

**CC 2: Comment from Ping Chen:**

**Reply:**

Thanks for the comments! During the reply, we found the page numbers and the reference numbers are not consistency to the manuscript, then based on the specific issues mentioned in these comments, we response as the following to clarify related data processing and analysis issues in the comments.

1. **On page 22, in the third step, what is the specific process of data screening based on the gamma parameter? How are Hs and kp in the empirical spectrum obtained? Are they directly calculated from the SWIM spectrum data?**

   **The Hs data of the SWIM spectrum is derived from the Hs at the nadir point.**

   **The SWIM spectrum data exhibits obvious spurious peaks and the surfbeat effect under low to medium sea states. Note that these are two different non - linear effects. Therefore, the spectrum data under such sea states is incorrect and cannot be used as reference spectrum data. Or the wave parameters kp and other conclusions obtained using these spectra as reference data are unreliable.**

   **Reply:** We appreciate the reviewer's valuable comments regarding the data screening process based on the gamma parameter. During the preprocessing SWIM L2 product data, we employed a 0.5%-99.5% confidence interval to eliminate abnormal samples of peak enhancement factors outside this range. Statistical analysis shows that this confidence interval corresponds to peak enhancement factor values ranging approximately from 1 to 12, so we ultimately selected wave units with peak enhancement factors between 1-12 as our research data.

   Regarding the construction of the Combined spectrum (C spectrum), we utilized both significant wave height $H_{\frac{1}{3}}$ and peak wavenumber $k_p$ information from SWIM L2 data.

   The significant wave height $H_{\frac{1}{3}}$ is directly provided by the SWIM L2 product, while the peak wavenumber $k_p$ can be calculated from the dominant wavelength $\lambda_p$ data in the SWIM L2 product.

   We consider that spurious peaks and the surfbeat effect may cause the observed spectral peak to correspond to low-wavenumber false peaks, leading to an underestimated maximum peak wavenumber $k_{max}$ and affecting data reliability. For this, we implemented quality control by calculating the difference $\Delta = k_p - k_{max}$ and setting a reasonable threshold to exclude samples with excessively large $\Delta$ values. This has been explained at the content around the last 3 lines in part 3.1.1. Specifically, in practice, $k_p$ is directly obtained from SWIM L2 data, while $k_{max}$ is determined by identifying the wavenumber corresponding to the maximum peak in the observed spectrum. Since false

peak data typically exhibit significant discrepancies between $k_p$ and $k_{max}$, whereas valid data should show close agreement between these values, $\Delta$ should theoretically be as small as possible. In this way, the data with those effects are screened out. Statistical analysis of the $\Delta$ distribution guarantees and verifies this quality control procedure, and shown specifically in the following figure.

[Figure]

(a)            (b)

Figure Re-CC2-1 statistic on $\Delta$

The figure above (Figure Re-CC2-1) shows the frequency histogram and cumulative probability density histogram of $\Delta$, showing that over 99.5% of the data samples are positive, indicating that most $k_p$ values lie to the right of $k_{max}$. The median $\Delta$ is 0.007, with a mean value of 0.016. Panel (a) reveals that the data are primarily concentrated between 0-0.007, and panel (b) demonstrates that this range accounts for half of the total samples (approximately 1.5 million boxes). To minimize data errors, we selected the median $\Delta$ as the threshold for screening, ensuring that all obtained data fall within this threshold.

To address this clearly, we update part 3.1.1 for description of SWIM data. We have expanded the content regarding data screening using the $\Delta$ threshold of the revised manuscript. The content has been modified from "Subsequently, to suppress abnormally high values at low wave numbers of SWIM caused by parasitic peaks (Merle et al., 2021; Xu et al., 2022), " to "To suppress abnormally high values at low wave numbers of SWIM caused by spurious peaks and surfbeat effects (Merle et al., 2021; Xu et al., 2022), Data screening was implemented by calculating $\Delta = k_p - k_{max}$ and setting a reasonable threshold to exclude samples with excessing large $\Delta$ values. $k_p$ represents the SWIM-observed peak wavenumber and $k_{max}$ corresponds to the wavenumber of maximum spectral energy. Since false peak data typically exhibit significant discrepancies between $k_p$ and $k_{max}$, whereas valid data should show close agreement between these values, $\Delta$ should theoretically be as small as possible. As shown in Fig. 1, the frequency histogram in panel A reveals that the data are primarily concentrated below 0.007, and the cumulative distribution in panel B demonstrates half of the total samples fall within this range for $\Delta \leq 0.07$. After balancing the trade-off between the sufficiency of retained samples and screening effectiveness, 0.007 was selected as the threshold value. Subsequently, according to the histogram of $\gamma$, the boxes for $\gamma$ ranging from 1 to 12 are selected as the research data that depicts the sea surface well. In

total, 1 251 067 boxes are found as the experiment data in 2021 for modelling, and 1 307 256 boxes in 2022 are obtained for validation." And Figure 1 below is newly included:

[Figure]

**Figure 1: The histogram and Cumulative Histogram of Δ**

2. **On page 33, in Figure 3 - 9, is the SWIM spectrum the average of multiple samples? Due to the statistical fluctuations of the SWIM spectrum, even under the same sea surface conditions, the kp corresponding to the SWIM - measured spectrum still varies. Directly averaging the spectra will lead to a decrease in the spectral peak and an increase in the spectral width. Therefore, the common practice is to perform wavenumber - normalized spectral averaging. A detailed description of this method can be found in Ref. [52].**

   **Reply**: Thanks to the reviewer for pointing the issue of the potential for direct averaging required for SWIM spectra. Spectral averaging under the same sea surface conditions may lead to reduced peak intensity and increased spectral width due to statistical fluctuations in SWIM spectra. Though we didn't demonstrate the SWIM spectra in figures, the averaging issue is a good point for SWIM data applications. Then we would like to clarify that for the development of the Combined spectrum. In this manuscript, we perform parameter fitting using combination of bunches of individual observation samples. Thus, instead of focusing on individual expression of the spectrum, we focus on the statistics features, this has been explained in the manuscript in part 3.1.1 and section 3.2 for data amount and data applied specifically in the fitting details respectively, and the concerned spectral averaging does not affect our results.

3. **Because the SWIM spectrum data has its own problems under low to medium sea states, it is not suitable to use SWIM data as reference data to establish an empirical spectrum model.**

   **Reply**: We appreciate the reviewer's comment regarding the limitations of SWIM spectrum data under low to medium sea states and its suitability as reference data for establishing an empirical spectrum model. We fully acknowledge that SWIM wave spectrum data may be affected by nonlinear effects such as spurious peaks and the surfbeat effect under these conditions. To minimize this adverse effect, we implemented

data clean primarily based on the difference between peak wavenumber $k_p$ and the maximum peak wavenumber $k_{max}$, as detailed in our reply to your first comment (CC2 Comment 1), to exclude abnormal wave box data as much as possible with mainly data expressing the surface features obtained for our further procedures. Moreover, for validation of our results, in addition to using SWIM observation data, we also employed independent validation with NDBC buoy data, as in part B of section 3.4.1. Evaluation results using both DI and $R^2$ metrics demonstrate that compared to classical wave spectrum models such as Elfouhaily and Goda spectra, the Combined spectrum can more accurately characterize real sea conditions. This also in another aspect, validated the method proposed. With the response to the first comment, we updated part 3.1.1, by inclusion of specific details for date selection, the specific update please refer to the reply to your comment 1 (CC2 Comment 1).

---

## Author Response (AR1)

**Dear Editor and Reviewers,**

Thanks for your work and precious comments on this manuscript! We 've made a thorough consideration and modification accordingly. We have highlighted the changes in the manuscript content. Here is a explain of the author information and a point-by-point response to the comments from CC and RC in time order, follow by other changes made accordingly to number of indexes of equations and figures, as well asl in following suggestions of the reviewer to check the English expressions that not list specifically in the comments. Other minor changes, such as deleting additional space, are not listed, and please refer to the submitted revision with trackable mode.

During the revision, Yihui Wang has graduated in July of 2024, with the change of her author information explained in this submitted manuscript: "The research contributions of Yihui Wang to this paper have been completed at the affiliation indicated by the authors. Her current affiliation is as follows: CACES Technologies (Hang Zhou) Co.,Ltd.". Meanwhile, Ziming Dong is now taking roles in this research as duplicating the existing experiments and performing addition experiments necessary for replying reviewers' comments for this research, thus included as another author for this manuscript in this revision.

**The comments from the reviewers:**

**CC 1: Comment from Wenming Lin:**

This paper combines the widely used Goda and elfouhaly spectra in order to improve the one-dimensional wave spectrum, The SWIM data are used for both model development and result verification. The NDBc buoy data is used for validation aswell. Based on the evaluation of Dl and  $R^2$  parameters, it shows that the proposed spectrum (C spectrum) generally performsbetter than G/E spectrum in term of characterizing the wave energy distribution. The contents of this paper are clear, and the steps of the methodologies are well described. The manuscript is generally well written, though the English is not native and needs polishing.

**Reply:** Thanks a lot for your precious comments on this manuscript! We've improved the expression in this version and put the explanation of the changes after the replies to specific comments:

1. Title: "An Improved One-dimensional Ocean Wave Description based on SWIM Observations" may be more concise.

**Reply:** Thanks for your suggestion. We have modified the title.

**2. Structure:**

a). Keep concise on the description of G/E spectra, and highlight the C spectrum. Particularly, please clarify how you fit the (23)-(24) and (26) – (28).

**Reply:** The structure of the C-spectrum is a fusion of the partial structures of the E-spectrum and G-spectrum. We have now put the formulas and physical meanings of the C spectra related to the adopted from the G spectrum and E spectrum in the C spectrum description to make the description of C spectrum concise. And for G and E spectra, we keep necessary descriptions. In detail, we modified the  $L_{PM}$  in Eq. (3),  $J_P$  in Eq. (4), and Lim in Eq. (15) in the original version to Eq. (11), Eq. (13), and Eq. (18) in the revised manuscript, and the structure of the relevant text is also modified. Please see specifically the modification in the change-trackable manuscript.

For the fitting of equations mentioned in the comment, in order to study the relationship between spectral peak enhancement factor  $\gamma$ , wave steepness and inverse wave age, the data were divided into several subsets according to wave steepness with different binning numbers for with references to the inverse wave age. The wave steepness of the wave measurements are mainly distributed in the region of 0.004-0.0115, and for the SWIM measurement in this region, there are 41 bins with 0.0002 wave steepness and 0.04 inverse wave age as the set intervals. The rest of the measurements were divided with 19 bins in the wave steepness interval 0.001 and inverse wave age interval 0.04. The mean values of inverse wave age, wave steepness and  $\gamma$  in each grid were calculated respectively. For the calculation of  $\gamma$ -mean value, the number of wave cells in the set binning was counted at first, and it was considered that the statistical characteristics are weak when the binned grid is with less than 30 wave cells, where the  $\gamma$ -mean value is set to 0 and not applied in the regression, if the number of wave elements in the grid is more than 30, the  $\gamma$  confidence interval is set to be 0.5%  $\sim$  99.5%, and the  $\gamma$  of the confidence interval is averaged for its value in the fitting.

Because the curve shapes of  $\gamma$  and inverse wave ages corresponding to the steep waves are closer to the trigonometric function, the Fourier series was chosen to fit the  $\gamma$  and inverse wave age in each steep interval.

At last, based on an appropriate function form, the coefficients a0, a1,a2 were fitted to the wave steepness.

b). in the result section, you may present the case study (3.4.2) firstly and then illustrate the general verifications.

**Reply**: Thanks for the suggestion, however, the case is provided for a concrete view of the results based on the validated theory, then the verification and validation of the proposed theory

is illustrated before the case, They are provided for convincing of the validation as well. We would like to keep the current order if it is also okay.

3. Regarding the combination of C spectrum, it's not a surprise that C spectrum generally agrees better with G spectrum than E spectrum. Is there any case showing that the old G/E spectra may be superior to C spectrum? Why?

**Reply**: Thank you for this comment, according to which, we make some new illustration and implement as new input of discussions in the conclusions part. Where we find cases that the G/E spectrum may be superior to the C spectrum for analysis.

Specifically, we find such data set with R-square index. Then the statistics in terms of density distributions of inverse wave age and wave steepness in the three cases: 1) the G spectrum is better than the C spectrum, 2) the E spectrum is better than the C spectrum, and 3) the C spectrum is better than the G and E spectra are investigated. During the procedure, we adopt the sea state classification metrics used in Section 2.1 of the reference article(Hauser et al., 2009; Xu et al., 2022; Hwang, 2009), which identifies measurements

at  $\Omega < 0.84$  and  $\delta > \frac{2\sqrt{3.64 \times 10^{-3}}}{\pi} \Omega^2$  as a mixed sea state with swell-dominated feature,

and  $\Omega > 0.84$  and  $\delta \leq \frac{2\sqrt{3.64\times10^{-3}}}{\pi}\Omega^2$  are as younger wind wave sea states, while

 $\Omega>0.84$  and  $\delta>\frac{2\sqrt{3.64\times10^{-3}}}{\pi}\Omega^2$  are classified as mixed sea states in which wind wave are dominant. It can be observed in the following figure where inverse wave age and wave

steepness distributions are taken as measurement of the sea evolving features obtained with

 $R^2$  as reference, and the orange curve in the figure represents  $\delta = \frac{2\sqrt{3.64\times10^{-3}}}{\pi}\Omega^2$ , the red dashed line represents  $\Omega = 0.84$  and the black dashed line represents  $\Omega = 1.0$ . That most measurement in (a) where G spectra is superior to C spectra is located in the region of swell-dominated mixed sea state. Meanwhile in (c), which illustrates the cases where G is superior to C according to  $R^2$ . This suggests that the G spectrum outperforms the C spectrum in the more mature swell sea state. As in the following figure:

Figure Re-CC2-1. Sea state distribution when the old G/E spectrum outperforms the C spectrum, using  $R^2$  as index for obtaining: as the validation index. (a) the G spectrum is better than the C spectrum, (b) the E spectrum is better than the C spectrum and the E spectrum.

And when comparing (b) with (c), it can be found that the regions with higher density in t(b) are mainly concentrated in two regions distributed near the parabola, one with smaller inverse wave age and the other with larger inverse wave age, which demonstrates that the E spectrum perform better than the C spectrum most in steepness larger than 0.025 while inverse wave age larger than about 0.8, in accordance to younger wind-wave dominated mixed sea states and wind-wave sea states. Corresponding content is included in the discussion of this submission. Thanks again for your comment.

**Minor comments:**

1. Do you fit the model using SWIM data, and then calculate R2 and DI for the same wavelength bin as SWIM? Please clarify the effective range of SWIM wavelength in the text, as well the range of integration in Eqs. (21) - (23).

**Reply:** We fit the  $\gamma$  of the model using SWIM data. and calculate R2 and DI for the same wavelength bin as SWIM. The effective range of SWIM wavelength in the text is 30~600m (correspond on 0.01~0.2 rad/m) and the same wavelength used in the Eqs (21)-(23).

2. Page 1, lines 15-16: two "then" appears in the two sentences.

**Reply:** We have adopted the first "then" to "and" and modified accordingly.

3. Page 1, lines 22-23: "The DI and R2 for the C .... 0.909 respectively". It is not necessary to introduce the detailed numbers here. Lines 24-25, "Further research would ... directions." Remove this sentence.

**Reply:** We have removed the sentence "and values are 0.780 and 0.0909 respectively" in line 23 and "Further research would ... directions" In line 24-25 in our previous manuscript.

4. Introduction, there are many other wave spectra not reviewed in this section, such as Huang's model.

**Reply:** The Goda and Elfouhaily spectra are the major theoretical models used in the research. We have included more wave spectra, for example Hwang, and Kareav spectra in the revision in the introduction part for a better background according to this comment.

5. Page 2, lines 57-58, "validation of the C spectrum ... of the sea surface." Conclusion should go to the conclusion section.

**Reply**: Thanks for this suggestion. Anyhow, this is mainly an introduction to the data in the validation and we modified the sentence to clarify. So could we keep in this way.

6. Page 3, lines 73-74: "... describing the inverted transfer of wave ...". This sentence is vague, please rephrase.

**Reply**: This sentence is redundant and not related with the topic, we have removed it.

7. Page 5, lines 127-132. "In comparsion .... observations well." Are the descriptions the results of this manuscript or previous studies?

Reply: Thanks for this suggestion. This can be referred to a previous study (Wang et al., 2023), which is mentioned in the previous sentence of the original manuscript.

8. Page 7, line 185, "the Surface Waves Investigation and Monitoring (SWIM) carried". Not necessary to write abbreviation again here. BTW, which version of SWIM data is used in this study.

Reply: The version is OP06, and we have added the version in the text.

9. Page 8, lines 200 - 201, informal numbers in the text.

Reply: We have modified it according to your suggestion

10. You may use SWH instead of H1/3 in the text.

**Reply:** H1/3 expression is now mainly used for formula.

11. Page 11, figure 4, I don't see the variation of H1/3 in the caption or in this figure. Please clarify.

**Reply:** Eq. (18) in our manuscript illustrates the relationship between significant wave height  $H_{\frac{1}{3}}$  and wave steepness  $\delta$ , If the significant wave height changes and the wind

speed and spectral peak number are fixed, then the wave steepness changes, while the inverse wave age is a fixed value. The larger the significant wave height, the steeper the wave, and the larger the integral area of the curve. We have added detailed  $H_{\frac{1}{3}}$  legend in Figure 5 of the revised manuscript.

**12. Remove "This is" in the captions of Figures 7-10.**

**Reply**: We have removed the two words "This is" in the captions of Figure 7-10.

**13. Page 15, line 325, this sentence is vague, please rephrase.**

**Reply**:" Fig. 9 (a), most DI of the C height spectrum are lower than those of the G height spectrum, but the proportion of superiority is not as significant, and the C spectrum has a superiority proportion with DI concentrated in the region where y

Figure Re-CC2-1 statistic on  $\Delta$

The figure above (Figure Re-CC2-1) shows the frequency histogram and cumulative probability density histogram of  $\Delta$ , showing that over 99.5% of the data samples are positive, indicating that most  $k_p$  values lie to the right of  $k_{max}$ . The median  $\Delta$  is 0.007, with a mean value of 0.016. Panel (a) reveals that the data are primarily concentrated between 0-0.007, and panel (b) demonstrates that this range accounts for half of the total samples (approximately 1.5 million boxes). To minimize data errors, we selected the median  $\Delta$  as the threshold for screening, ensuring that all obtained data fall within this threshold.

To address this clearly, we update part 3.1.1 for description of SWIM data. We have expanded the content regarding data screening using the  $\Delta$  threshold of the revised manuscript. The content has been modified from "Subsequently, to suppress abnormally high values at low wave numbers of SWIM caused by parasitic peaks (Merle et al., 2021; Xu et al., 2022), "to "To suppress abnormally high values at low wave numbers of SWIM caused by spurious peaks and surfbeat effects (Merle et al., 2021; Xu et al., 2022), Data screening was implemented by calculating  $\Delta = k_p - k_{max}$  and setting a reasonable threshold to exclude samples with excessing large  $\Delta$  values.  $k_p$  represents the SWIM-observed peak wavenumber and  $k_{max}$  corresponds to the wavenumber of maximum spectral energy. Since false peak data typically exhibit significant discrepancies between  $k_p$  and  $k_{max}$ , whereas valid data should show close agreement between these values,  $\Delta$  should theoretically be as small as possible. As shown in Fig. 1, the frequency histogram in panel A reveals that the data are primarily concentrated below 0.007, and the cumulative distribution in panel B demonstrates half of the total samples fall within this range for  $\Delta \leq 0.07$ . After balancing the trade-off between the sufficiency of retained samples and screening effectiveness, 0.007 was selected as the threshold value. Subsequently, according to the histogram of  $\gamma$ , the boxes for y ranging from 1 to 12 are selected as the research data that depicts the sea surface well. In total, 1 251 067 boxes are found as the experiment data in 2021 for modelling, and 1 307 256 boxes in 2022 are obtained for validation." And Figure 1 below is newly included:

Figure 1: The histogram and Cumulative Histogram of  $\Delta$

2. On page 33, in Figure 3 - 9, is the SWIM spectrum the average of multiple samples? Due to the statistical fluctuations of the SWIM spectrum, even under the same sea surface conditions, the kp corresponding to the SWIM - measured spectrum still varies. Directly averaging the spectra will lead to a decrease in the spectral peak and an increase in the spectral width. Therefore, the common practice is to perform wavenumber - normalized spectral averaging. A detailed description of this method can be found in Ref. [52].

Reply: Thanks to the reviewer for pointing the issue of the potential for direct averaging required for SWIM spectra. Spectral averaging under the same sea surface conditions may lead to reduced peak intensity and increased spectral width due to statistical fluctuations in SWIM spectra. Though we didn't demonstrate the SWIM spectra in figures, the averaging issue is a good point for SWIM data applications. Then we would like to clarify that for the development of the Combined spectrum. In this manuscript, we perform parameter fitting using combination of bunches of individual observation samples. Thus, instead of focusing on individual expression of the spectrum, we focus on the statistics features, this has been explained in the manuscript in part 3.1.1 and section 3.2 for data amount and data applied specifically in the fitting details respectively, and the concerned spectral averaging does not affect our results.

3. Because the SWIM spectrum data has its own problems under low to medium sea states, it is not suitable to use SWIM data as reference data to establish an empirical spectrum model.

**Reply**: We appreciate the reviewer's comment regarding the limitations of SWIM spectrum data under low to medium sea states and its suitability as reference data for establishing an empirical spectrum model. We fully acknowledge that SWIM wave spectrum data may be affected by nonlinear effects such as spurious peaks and the surfbeat effect under these conditions. To minimize this adverse effect, we implemented

data clean primarily based on the difference between peak wavenumber  $k_p$  and the maximum peak wavenumber  $k_{max}$ , as detailed in our reply to your first comment (CC2 Comment 1), to exclude abnormal wave box data as much as possible with mainly data expressing the surface features obtained for our further procedures. Moreover, for validation of our results, in addition to using SWIM observation data, we also employed independent validation with NDBC buoy data, as in part B of section 3.4.1. Evaluation results using both DI and  $R^2$  metrics demonstrate that compared to classical wave spectrum models such as Elfouhaily and Goda spectra, the Combined spectrum can more accurately characterize real sea conditions. This also in another aspect, validated the method proposed. With the response to the first comment, we updated part 3.1.1, by inclusion of specific details for date selection, the specific update please refer to the reply to your comment 1 (CC2 Comment 1).

**RC1: Comment from Anonymous Referee #1:**

The manuscript proposes a combined one-dimensional spectrum of ocean surface waves, with wavelengths ranging from swell to wind wave scales, to improve measurement predictions compared to existing models. The model is built using SWIM satellite measurements as input data, and validation is performed using buoy data and a separate set of SWIM measurements. Some parameters and comparison results should be revised.

**Reply**: We sincerely appreciate the thorough review and constructive suggestions. Below are our point-by-point responses to the comments:

1. Page 5, lines 115 through 125: The authors compare G spectrum and E spectrum predictions of SWIM measurements and define specific wavenumber intervals where each estimation is in agreement with the observed data. Additional to that information, a visual inspection of the predictions against measurements would be helpful. While the method section describes the SWIM dataset (lines 185 - 200), it is unclear how the wavenumbers ranging from 31 m to 209 m were resolved. Providing a time series of SWIM measurements, along with sampling rate and Nyquist wavenumber and frequency, would clarify the analysis. Note that in the section 3.1.2 (line 205), the SWIM measurement range is stated as 0.01–0.2 rad/m-1, but this comes much after than the initial description of the SWIM dataset.

**Reply:**

Thanks for the suggestion! The different wavelengths of G and E spectra are determined by the detailed procedures of individual spectra, it would be lengthy to include the specific content of the existing literature on this, thus we put the features and cite the parameters of E and G spectra in references (Elfouhaily, T, et al, 1997) and (Goda, Y., 1983) for this manuscript. For SWIM measurement, we acknowledge the value of including radar time series, and we believe these instrument-specific details would be more appropriate in instrumentation-focused papers, as it would be impossible to clarify this in a few sentences and figures. Moreover, there is detailed introduction of how these measurements are made and inverted from measured NRCS in different incidence angles in the very detailed user manual, it would be also too long to include. Furthermore, inclusion of these content would make the manuscript somehow derive from its focus, especially in the method part, so following this suggestion, to make the description with more details, we include the following references introducing the principles with time variance (echo gates) and specific producing procedures of spectrum for readers who are interested to the data description section: (Hauser, D., Tison, C., Amiot, T., Delaye, L., Corcoral, N., and Castillan, P.: SWIM: The First Spaceborne Wave Scatterometer, IEEE Transactions on Geoscience and Remote Sensing, 55, 3000-3014, https://doi.org/10.1109/TGRS.2017.2658672, 2017.) and (CNES: SWIM Products User Guide, Technical Report CF-GSFR-MU-2530-CNES, AVISO+, 2023.). These references have been were added to the revised manuscript. Now the sentence in 3.1.1 has been modified from "SWIM products provide valuable wave spectrum details as well as wave parameter information in global coverage." to "SWIM version OP06 products provide valuable wave spectrum details as well as wave parameter information in global coverage, with

observed wavelengths ranging from 31 to 628 m (corresponding to wavenumbers from 0.01 to 0.2 rad.m-1) (Hauser et al., 2017; CNES, 2023)". In addition, we have now supplemented the key parameters of SWIM observations (Including a range of observation wavelengths from 31 to 628m) in the revised manuscript in section 2.1, the last second paragraph, from "For the SWIM measurements, they provide the descriptions of real sea that is by nature of different states. Although it is focusing on longer waves in the spectrum," to "The SWIM measurements observe the descriptions of real sea that is by nature of different states. It is focusing on longer waves in the spectrum, corresponding to wavelength from 31 to 628m.".

**2. Equation 20: It can be either written as $H_{1/3}$ or $H_s$ , it is redundant to use both. Reply: We have unified the notation for significant wave height as $H_{\frac{1}{3}}$ throughout the manuscript.**

3. Equation 23 and 24: The coefficient fit formulas do not match with their corresponding graphs on Figure 2; are those reversed?

**Reply:** Thank you for catching this error. We confirm that coefficients a0 and a1 were mistakenly reversed in Figure 2 of our prior manuscript. In the revised version, we've corrected the coefficient labels in the original figure. **There is a new figure included from comments of aother reviewer, Figure 2 is now Figure 3,** with the left panel is now showing the correct coefficient a0 and the right panel the correct coefficient a1):

Figure 3: The fitness of  $a_0$ ,  $a_1$  with the wave steepness

**4. Line 234: Wave steepness range should be 0.0115, not 0.015.**

**Reply**: Thanks a lot for the correction! We have corrected the wave steepness range to 0.0115 in this sentence.

**5. Line 249: The variable should be denoted as $k_p$ , not kp.**

**Reply**: The notation has been corrected to  $k_p$  and we've verified this notation throughout the manuscript.

**6. Line 252: What is the exact definition of a height spectrum? Is it wave height spectrum or wave elevation spectrum which is H/2?**

**Reply**: This refers to the wave height spectrum. For better clarifying, we modified the sentence in the second paragraph of section 3.3 from "Fig. 5 (a) $\sim$ (c) depict respectively the C spectrum in height spectrum," to "Fig. 5 (a) $\sim$ (c) depict respectively the C spectrum in wave height spectrum (height spectrum),"

7. Line 256 with relation to the Figure 4: It is written that "the trend of change is almost identical to the G spectrum, with the spectral peak in the C spectrum slightly smaller than that in the G spectrum" but the G spectrum results are not demonstrated.

**Reply**: Considering that giving the G-spectrum and the associated description may lead to too much redundancy, we chose to cite the trend analysis of the G-spectrum changes given in Fig. 4 of a previous study (Wang et al., 2023), here specifically shown below:

Goda Wave Height Spectrum

Goda Wave Slope Spectrum

Goda Wave Curvature Spectrum

Comparison with the C spectrum (**Figure 5 now**) reveals that the variation trend of the C spectrum is almost identical to with that of the G spectrum in terms of the height and slope spectra, though the spectral peak intensity of the C spectrum is slightly lower than that of the G spectrum. In the curvature spectrum, the values of the C spectrum on the right side of the spectral peak gradually increase, contrasting with the trend in the G spectrum where the curve gradually flattens

8. The Figure 4 caption describes spectrum results at different significant wave heights, while the figure legends display wave steepness values and wave age. Although wave steepness is directly related to wave height, the caption and legend should use the same parameter for clarity. Additionally, since wave age remains constant throughout Figure 4, it does not need repetition in each legend entry.

**Reply**: Thanks for the suggestion. We have unified the presentation to use significant wave height consistently in both caption and legends, and removed redundant wave age information. Now **Figure 5** of the revised manuscript:

Figure 5: The C spectrum at different  $H_{\frac{1}{3}}$ , when  $k_{\rm p}$  is 0.048 rad·m-1 and  $u_{10}$  is 10 m·s-1, (a) is Combined height spectrum, (b) is Combined slope, (c) is Combined curvature.  $\delta$  is wave steepness, and  $\Omega$  is inverse wave age.

9. Figure 5: The previous comment applies to this figure as well. Both wave age and wave steepness change here due to variations in peak wavenumber, not wind speed. For clarity, the legend should show only the varying parameter (peak wavenumber or wave steepness) rather than listing all dependent variables such as wave age.

**Reply**: In a varied sea state, the parameters are linked with each other. For conciseness, following this suggestion, we now use peak wavenumber as the sole varying parameter in the legend. It is now **Figure 6** of the revised manuscript:

Figure 6: The C spectrum at different  $k_p$ , when  $H_{\frac{1}{3}}$  is 3 m and  $u_{10}$  is 10 m·s-1, (a) is Combined height spectrum, (b) is Combined slope, (c) is Combined curvature

10. Figure 6: The only varying parameter is wind speed yet the legend does not indicate any wind speed variation. Unlike Figure 5, where wave ages vary due to peak wavenumber changes, here they vary due to wind speed changes. The figure should clearly indicate which parameter is being varied.

**Reply**: Thanks for this good suggestion to make the figures clearer! We have modified the figure to clearly indicate wind speed as the varying parameter. Now **Figure 7** of the revised manuscript:

Figure 7: C spectrum at different wind speed, when significant wave height is 3m and  $k_{\rm p}$  is 0.048 rad·m-1, (a) is Combined height spectrum, (b) is Combined slope, (c) is Combined curvature

11. Figure 8a:It is hard to see the probability density variations. the colorbar for all plots of figure 8 should be in similar range so that we can better see the comparison.

**Reply**: We have adjusted the colorbars to use similar ranges for the same metrics (DI metrics in figure 8a and 8c and R2 metrics in figure 8b and 8d in our revised manuscript), with optimized ranges to better show density variations. In addition, we also made the similar modifications to Figure 7 of the revised manuscript, they are now Figure 8 and 9 in the modified manuscript:

Figure 8: Validation of DI and R2 Metrics from C and E Spectra with SWIM in 2022, (a), (c), is DI of height and curvature spectrum, (b), (d) is R2 of height and curvature spectrum

Figure 9: Validation of DI and R2 Metrics from C and E Spectra with SWIM in 2022, (a), (c), is DI of height and curvature spectrum, (b), (d) is R2 of height and curvature spectrum.

**12. The headings of Table 1: "The C spectrum better than the G/E spectrum" is unclear. What does DI and R values for C spectrum better than E spectrum mean?**

**Reply**: We have revised the table to clearly indicate that the values represent the percentage of cases where the C spectrum outperforms the G/E spectrum across different evaluation metrics (DI of curvature/height and R2 of curvature/height). For example, Combined with the "DI of curvature" row of the table, we can understand that The proportion of curvature and height spectrum in DI about the C spectrum better than the G spectrum is 82.9% and 78.0%, which means the C spectrum fits more the SWIM measurements. Specifically the content in Table 1has been modified from "The C spectrum better than the G spectrum" to "Percentage of the C spectrum better than the G spectrum".

**13. Table 2: The previous comment applies to this table as well.**

**Reply**: Similar to Table 1, we have modified the presentation to prevent misinterpretation. Specifically the content has been modified from "The C spectrum better than the G spectrum" to "Percentage of the C spectrum better than the G spectrum".

**14.** Figure 11: Height spectra (Elevation spectra?) y label units are missing. Is it normalized?**

**Reply**: Thanks for the correction, it should be clarified that no normalization has been operated here. We have added the missing y label units to Figures 11-and 14 (Now 12 and 15) in the revised manuscript:

Figure 12: Comparison of the SWIM, G, E and Combined height spectra when wave steepness is 0.02, and inverse wave age is 0.479.

Figure 15: Comparison of the SWIM, G, E and Combined curvature spectra when wave steepness is 0.026, and inverse wave age is 0.803.

**RC2: Comment from Anonymous Referee #2:**

Publisher's note: this comment was edited on 6 May 2025. The following text is not identical to the original comment, but the adjustments were minor without effect on the scientific meaning.

Thanks for the note! We hereby provide responses to the newest version of comments.

Comments: This is a very interesting work with results showing the benefit of using directional wave spectra from wave scatterometer SWIM of CFOSAT. The paper focuses on the development of a parametric wave spectrum that includes an enhancement parameter depending on sea state conditions such as wave steepness, wave age, and wind. The use of L2 SWIM spectra allows for the adaptation and calibration of this enhancement parameter. Validation of the established combined spectrum, in comparison with other parametric wave spectra (Goda and El-fouhaily) and NDBC buoys and SWIM observations, demonstrates more consistency with observations from C spectrum and opens very interesting perspective of using the combined spectrum for remote sensing applications. The analysis is well detailed, taking into account the variability of parameters describing sea state. I would recommend to authors to outline in the conclusions a room of improvement related to using wind speed from observations instead of model's one, which include uncertainties in some key ocean regions.

The paper is well-written, and the sections are clearly defined. Below are some comments to improve reading the text and also points to clarify:

**Reply:**

We sincerely appreciate the positive feedback and constructive suggestions for improving our manuscript from the reviewer. We have included a sentence in the conclusion following this suggestion, as the last sentence in the last paragraph has been modified from: "...and is our further research based on the established C spectrum." to "...and is our further research based on the established C spectrum, when the collocated measured wind will be applied for better characterize the observed scene".

Below are responses to the specific comments:

**1. figure 4 a, b and c are not indicated on figures, you can just mention left, middle and right**

Reply: Thanks a lot for this correction. To keep consistency with other figures with subplots, we have added (a), (b), and (c) labels to each panel of Figure 4 in the previous manuscript(Figure 5 in the revised manuscript, as one figure has been included according to the comments of another reviewer). From here we also notice the same problems exist for Figure 2 and Figure 3 in our previous manuscript, the sub-plots are also labeled accordingly. (Figure 3 and Figure 4 in the revised manuscript)

**2. figure 7 & 8 : more comprehensive title, for instance validation of $R^2$ and DI from C and G spectra.**

**Reply**: Following this suggestion, we have revised the titles of Figure 7 and Figure 8. As the figure number has also been changed due to inclusion of another figure according to another reviewer's comment, the annotations are changed from: "Figure 7: This is the comparisons of the validation index of E spectrum with C spectrum, (a), (c), (e) is DI of height, curvature and slope spectrum, (b), (d),(f) is R2 of height, curvature, and slope spectrum." and "Figure 8: This is the comparisons of the validation index about G spectrum with C spectrum with SWIM in 2022, (a), (c) is DI of height and curvature spectrum, (b), (d) is R2 of height and curvature spectrum." to: "Figure 8: Validation of DI and R2 Metrics from C and E Spectra with SWIM in 2022, (a), (c), is DI of height and curvature spectrum, (b), (d) is R2 of height and curvature spectrum " and "Figure 9: Validation of DI and R2 Metrics from C and E Spectra with SWIM in 2022, (a), (c), is DI of height and curvature spectrum, (b), (d) is R2 of height and curvature spectrum."

Similar modifications were also made to the previous Figures 9 and 10, The titles are changed to "Figure 10: Validation of DI and R2 Metrics from C and G Spectra Comparisons of the validation index of G spectrum and C spectrum with buoys, (a), (c) is DI of height and curvature spectrum, (b), (d) is R2 of height and curvature spectrum." And "Figure 11: Validation of DI and R2 Metrics from C and E Spectra with buoys, (a), (c) is DI of height and curvature spectrum, (b), (d) is R2 of height and curvature spectrum."

**3. Line 330: there is a typo on this line "better than G spectrum" I guess.**

**Reply**: Thanks again for pointing this. We have corrected the typo. Now this sentence has been changed from "Results indicate the C spectrum has a good fit with NDBC buoy measurements than the C spectrum." to "Results indicate the C spectrum has a better fit with NDBC buoy measurements than the G spectrum.".

**4. Figure 9 & 10: same remarks as for figure 7 and 8, you can remove the "this is the comparison", put just the title.**

**Reply**: We have simplified the titles as suggested, removing "This is" in the annotation of Figure 7, 8, 9 and 10 of the previous manuscript to make them more concise. See also the reply to comment 2 here from RC2.

**5. Line 420: rephrasing "higher proportion superiority"**

,,

**Reply**: This sentence has now been modified from "The evaluation indexes of the C spectrum model show the highest proportion of superiority, indicating that the C spectrum is closer to the measured data compared to the G and E spectra." to: "The evaluation indices of the C spectrum model show the superior performance, indicating that the C spectrum is closer to the measured data compared to the G and E spectra."

**Other modifications:**

**In addition to the changes mentioned above, there are other unmentioned changes summarized below:**

- 1) On page 5, line 94 of the revised manuscript, We have adjusted the writing of this paragraph to make the meaning clearer.
- 2) In section 3.4.1, subsection B, the subsection name has been modified, We replace "measurements" with "references".
- 3) Removed "This is" from the name of Table I.
- 4) The "Author contributions" section has been supplemented.
- 5) Minor revision to the last sentence of "Conclusions and discussions", advancing the words "In this research" to the end of the sentence.
- 6) Removed the last sentence in the abstract "Further research would .... Azimuthal direction"

**7) Modification to the figures:**

We have added a new figure before all the figures in our previous manuscript, therefore the number of all the figures has changed:

| previous manuscript's | revised manuscript's  |
|-----------------------|-----------------------|
|                       | Figure 1              |
| Figure 1 to Figure 15 | Figure 2 to Figure 16 |

**8) Modification to the equations:**

After the modification, the changes in the positions of some equations are shown in the following table:

| previous manuscript's | revised manuscript's |
|-----------------------|----------------------|
| Eq. (3)               | Eq. (11)             |
| Eq. (4)               | Eq. (13)             |
| Eq. (5)               | Eq. (15)             |
| Eq. (6)               | Eq. (3)              |
| Eq. (7)               | Eq. (4)              |
| Eq. (8)               | Eq. (5)              |
| Eq. (9)               | Eq. (6)              |

| Eq.(10)  | Eq. (17) |
|----------|----------|
| Eq. (11) | Eq. (12) |
| Eq. (12) | Eq. (7)  |
| Eq. (13) | Eq. (14) |
| Eq. (14) | Eq. (16) |
| Eq. (15) | Eq. (18) |
| Eq. (16) | Eq. (8)  |
| Eq. (17) | Eq. (9)  |
| Eq. (18) | Eq. (10) |

In addition, in the "Conclusions and discussions" section of the revised manuscript, Eq. (31) and Eq. (32) have been added, equations not mentioned above have not been modified.